# Exogenous Easily Extractable Glomalin-Related Soil Protein Stimulates Plant Growth by Regulating Tonoplast Intrinsic Protein Expression in Lemon

**DOI:** 10.3390/plants12162955

**Published:** 2023-08-15

**Authors:** Xiao-Niu Guo, Yong Hao, Xiao-Long Wu, Xin Chen, Chun-Yan Liu

**Affiliations:** 1College of Horticulture and Gardening, Yangtze University, Jingzhou 434025, China; 2022710887@yangtzeu.edu.cn (X.-N.G.); 2022710872@yangtzeu.edu.cn (X.-L.W.); 2022710868@yangtzeu.edu.cn (X.C.); 2College of Urban Construction, Yangtze University, Jingzhou 434023, China; 518004@yangtzeu.edu.cn

**Keywords:** aquaporin, water, root, nutrient, EE-GRSP

## Abstract

Arbuscular mycorrhizal fungi (AMF) have the function of promoting water absorption for the host plant, whereas the role of easily extractable glomalin-related soil protein (GRSP), an N-linked glycoprotein secreted by AMF hyphae and spores, is unexplored for citrus plants. In this study, the effects on plant growth performance, root system characteristics, and leaf water status, along with the changes of mineral element content and relative expressions of tonoplast intrinsic protein (*TIP*) genes in lemon (*Citrus limon* L.) seedlings, were investigated under varying strengths of exogenous EE-GRSP application under potted conditions. The results showed that 1/2, 3/4, and full-strength exogenous EE-GRSP significantly promoted plant growth performance, as well as increased the biomass and root system architecture traits including root surface area, volume, taproot length, and lateral root numbers of lemon seedlings. The four different strengths of exogenous GRSP displayed differential effects on mineral element content: notably increased the content of phosphorus (P) and iron (Fe) in both leaves and roots, as well as magnesium (Mg) and zinc (Zn) content in the roots, but dramatically decreased the content of calcium (Ca) and manganese (Mn) in the roots, as well as Zn and Mn in the leaves. Exogenous EE-GRSP improved leaf water status, manifesting as decreases in leaf water potential, which was associated with the upregulated expressions of tonoplast intrinsic proteins (*TIPs*), including *ClTIP1;1*, *ClTIP1;2*, *ClTIP1;3*, *ClTIP2;1*, *ClTIP2;2*, *ClTIP4;1*, and *ClTIP5;1* both in leaves and roots, and *TIPs* expressions exhibited diverse responses to EE-GRSP application. It was concluded that exogenous EE-GRSP exhibited differential responses on plant growth performance, which was related to its strength, and the effects were associated with nutrient concentration and root morphology, especially in the improvement in water status related to *TIPs* expressions. Therefore, EE-GRSP can be used as a biological promoter in plant cultivation, especially in citrus.

## 1. Introduction

Glomalin, secreted by the spores and hyphae of arbuscular mycorrhizal fungi (AMF) [1,2], and originally found by Wright and Upadhyaya [3], was demonstrated as a unique nitrogen (N)-linked glycoprotein by gel electrophoresis and sequencing [3]. Subsequently, researchers found that glomalin is a compound derived from soil; therefore, Rilling et al. [4] renamed it to glomalin-related soil protein (GRSP). With the continuous deepening of research, Wu et al. [5] proposed that GRSP should be divided into two predominant fractions, a newly produced fraction by AMF hyphae and spores that is easily extractable, which is called EE-GRSP; and an older glomalin that originated from the turnover of EE-GRSP and is difficultly extractable, called DE-GRSP.

Previous studies have shown that exogenous GRSP application could significantly promote plant growth. In citrus, exogenous EE-GRSP application significantly enhanced plant growth performance and the promotion effect was closely dependent on its strengths [6]. Subsequently, Chi et al. [7] selected 1/2-strength exogenous EE-GRSP and applied it to potted trifoliate orange seedlings under drought stress, and they found that this strength of exogenous EE-GRSP application significantly promoted the growth performance and biomass of the plants and alleviated the water stress, which was associated with the increased leaf water potential and the expression of ABA and IAA, etc., thus promoting the water uptake. In addition, the effects of EE-GRSP on plant growth and water uptake may also be related to the improvement in soil structure, which indirectly affects water and nutrient status [8]. GRSP acts as a coating on fungal hyphae, preventing loss of water, and the pronounced ability of GRSP to distribute on the soil aggregate surface as a hydrophobic layer also contributed handsomely to reduce water loss from and within soil aggregates, thus creating a very favorable relationship between soil water and plants [9].

The effective utilization of water by plants is momentous for its normal growth [10]. Aquaporins (AQPs) are the main channels for the active transport of water molecules on cells. These membrane-intrinsic proteins play a crucial role in water regulation, including maintaining the cell expansion, root water uptake, leaf transpiration, and response to environmental stresses [11]. AQPs are divided into seven subfamilies in terrestrial plants, with tonoplast intrinsic proteins (*TIPs*) being one of the main types [12]. *TIPs* belong to the tonoplast membrane intrinsic protein and play a decisive role in mediating water exchange between the cytoplasm and vacuole and alleviating cytoplasmic osmotic fluctuations [13]. Previous studies have revealed that *TIPs* are able to form a water flow in immature vascular cells and maintain permeability in mature vascular tissues, which is possibly dependent on the strong hydraulic properties of *TIPs* between the plasmalemma and tonoplast [14]. Earlier studies have reported that mycorrhiza-modulated water utilization thus improved the drought tolerance of host plants which was closely associated with AQP expression, such as in *Medicago truncatula* [15], soybean and lettuce [16], *Trifolium alexandrium* [17], and trifoliate orange [18]. However, whether GRSP, a glycoprotein secreted by AMF hyphae, can improve plant water status by regulating the expression of AQP remains unknown, and the research of GRSP on the water absorption of plants is scarce.

Citrus is an important fruit tree cultivated in more than 140 countries worldwide [19]. Lemon (*Citrus limon* L.) is an evergreen fruit tree belonging to *Citrus* of the rutaceae family and is widely used for its high edible and medicinal value. Although the effect of EE-GRSP on plant growth performance has been studied [6,7], little information is known about the underlying mechanism, especially in terms of water and nutrient absorption and transportation. In this study, lemon was selected as the experimental material to investigate the effects of five different strengths of exogenous EE-GRSP on water and nutrient uptake, plant growth performance, mineral elements, as well as the expression of *TIPs*, in order to clarify the physiological mechanisms by which GRSP improves water conditions and promotes plant growth.

## 2. Results

### 2.1. Growth Status and Biomass

Compared with the no-EE-GRSP citrate buffer control, the plant height and fresh weight in leaves and stems of lemon seedlings were significantly increased by 11.64~70.07%, 46.15~335.90%, and 52.63~210.53%, respectively, under four different strengths of exogenous EE-GRSP applications (Table 1). Lemon stem diameter, leaf number, and root fresh weight significantly increased under 1/2, 3/4, and full-strength EE-GRSP applications, by 23.10~32.67%, 55.81~60.47%, and 69.81~111.32%, respectively (Table 1).

### 2.2. Root Morphology

Root surface areas were significantly increased by 14.5% and 11.28% in 3/4 and full-strength exogenous EE-GRSP applications, respectively, as compared with the no-EE-GRSP citrate buffer control. Meanwhile, root volume, taproot length, and the number of 1st-order and 2nd-order lateral roots were significantly increased by 91.71~120.93%, 32.14~100.00%, 52.23~64.97%, and 188.82~377.64% under 1/2, 3/4, and full-strength exogenous EE-GRSP applications, respectively (Table 2). However, 1/4, 1/2, and 3/4 EE-GRSP conditions all dramatically decreased the number of 3rd-order lateral roots by 50.00%, 45.71%, and 50.00%, respectively, as compared with the no-EE-GRSP citrate buffer control. Moreover, 1/4-strength exogenous EE-GRSP application dramatically reduced the projected area by 13.66%, whereas all the four strengths of EE-GRSP applications did not affect the total root length and average diameter in lemon seedlings (Table 2).

### 2.3. Leaf Water Status

Compared with the no-EE-GRSP citrate buffer control, four different strengths of exogenous EE-GRSP applications did not significantly affect RWC (Figure 1a), whereas the leaf water potential of lemon seedlings were dramatically decreased by 54.55%, 63.64%, 81.82%, and 50.00% under 1/4, 1/2, 3/4, and full-strength exogenous EE-GRSP applications, respectively (Figure 1b).

### 2.4. Mineral Element

In lemon leaves, 1/4, 1/2, 3/4, and full-strength exogenous EE-GRSP applications substantially increased P content by 70.30%, 241.68%, 338.52%, and 214.12%, and Fe content by 19.51%, 19.86%, 84.53%, and 63.54%, respectively, but evidently decreased the content of Mn and Zn by 28.20%, 21.47%, 31.38%, and 29.81%, and 32.00%, 24.00%, 40.00%, and 44.00%, respectively, compared to no-EE-GRSP citrate buffer control (Table 3). In addition, N content was dramatically reduced by 47.49%, 44.57%, and 43.78% under 1/2, 3/4, and full-strength exogenous EE-GRSP applications, respectively. Interestingly, the contents of K, Ca, and Mg were not affected by the four strengths of exogenous EE-GRSP applications (Table 3).

In root, 1/2, 3/4, and full-strength exogenous EE-GRSP applications, the contents of P were prominently increased by 30.72%, 45.34%, and 14.75%; those of Mg increased by 41.93%, 54.68%, and 36.64%; those of Fe increased by 16.79%, 79.26%, and 35.33%; and those of Zn increased by 114.12%, 150.00%, and 23.13%, respectively, as compared with the no-EE-GRSP citrate buffer control, and the highest contents of these mineral elements were under 3/4-strength exogenous EE-GRSP application (Table 3). Furthermore, all four different strengths of exogenous EE-GRSP applications dramatically decreased Ca and Mn contents but did not affect K content (Table 3).

### 2.5. Relative Expression of TIPs

The result showed that in the four different strengths of exogenous EE-GRSP applications, only 1/4-strength exogenous EE-GRSP application significantly up-regulated the expressions of *ClTIP1;1*, *ClTIP1;2*, *ClTIP1;3*, *ClTIP2;1*, *ClTIP2;2*, *ClTIP4;1*, and *ClTIP5;1* in both lemon roots by 22.09-fold, 35.83-fold, 8.08-fold, 17.00-fold, 52.49-fold, 59.59-fold, and 76.55-fold, and in leaves by 111.34-fold, 3.05-fold, 1.96-fold, 9.96-fold, 10.83-fold, 4.36-fold, and 6.58-fold, respectively, as compared with the no-EE-GRSP citrate buffer control (Figure 2).

In addition, compared with the no-EE-GRSP citrate buffer control, 3/4-strength exogenous EE-GRSP application significantly up-regulated the expression of *ClTIP1;1*, *ClTIP1;3,* and *ClTIP2;1* in lemon roots by 1.60-fold, 6.64-fold, and 36.82-fold, and of *ClTIP1;2* and *ClTIP2;2* in lemon leaves by 1.43-fold and 1.84-fold, respectively. Meanwhile, the 3/4 strength sensibly down-regulated the expressions of *ClTIP1;3*, *ClTIP2;1*, *ClTIP4;1*, and *ClTIP5;1* in leaves and *ClTIP1;2*, *ClTIP2;2,* and *ClTIP5;1* in roots of lemon seedlings. The 1/2-strength exogenous EE-GRSP only significantly up-regulated the expression of *ClTIP2;2* in the leaf by 1.63-fold but dramatically down-regulated the other *ClTIP* expressions in both leaf and root. Full-strength exogenous EE-GRSP application undoubtedly down-regulated all the *ClTIP* expressions in lemon seedlings (Figure 2).

## 3. Discussion

The present study showed that exogenous EE-GRSP application significantly promoted the plant growth performance of lemon seedlings, which is consistent with the previous study by Wang et al. [6] and Liu et al. [20] on trifoliate orange seedlings. In addition, previous studies also revealed that the effect of exogenous EE-GRSP application on citrus growth was related to the application strength, and 3/4-strength exogenous EE-GRSP application showed the greatest positive effect on plant growth, whereas 1/2-strength exogenous EE-GRSP applications displayed the best performance in facilitating plant biomass [20]. The results of this study are in agreement with the above research findings. The ability of GRSP to promote plant growth is likely related to the presence of humic acid, which has been shown to improve the growth performance of various plants such as pea [21] and maize [22]. Moreover, Schindler et al. [23] also discovered that GRSP is composed of a variety of substances including aromatic hydrocarbons (42~49%), carboxylic acid groups (24~30%), carbohydrates (4~16%), and low aliphatic and carbon (4~11%), which are extremely similar to the structure of humic acid.

The root system architecture (RSA) is essential to plant water and nutrient utilization, which can be modified under environmental stress to enhance the nutrient-acquisition capacity [24]. It is well known that AMF colonization can ameliorate the RSA of host plants by promoting root hair growth and facilitating lateral root development [25]. GRSP, commonly known as an N-linked glycoprotein released by spores and mycelia of AMF, has also been found to maintain a greater root configuration parameter [7,20]. In this experiment, higher-strength exogenous EE-GRSP applications significantly increased lemon root project area, volume, and lateral root numbers, which is similar to the results of Liu et al. [20] on trifoliate orange seedlings. Moreover, the root average diameter of lemon seedlings was not affected by exogenous EE-GRSP application, which may be related to the growth and development pattern of the plant RSA. In general, the root system elongates radially before thickening laterally, thus ensuring an adequate supply of water and nutrients in the growth prophase and enhancing resistance in the growth anaphase [26].

Water is the decisive factor for the plants’ normal growth, and leaf RWC and water potential are important indicators reflecting the water status in plants [27]. Generally, RWC and water potential are positively correlated with the amount of intracellular water content [28]. However, in the present study, the four strengths of exogenous EE-GRSP applications did not affect the RWC but maintained it at high levels (92.8~94.8%), possibly as the environment where lemon seedlings grew had a suitable water condition, and the plants absorbed enough moisture from the surroundings, resulting in a higher water content in lemon seedlings. However, leaf water potential was significantly reduced after four different strengths of exogenous EE-GRSP applications, which could be attributed to an increase in solute concentration [29]. In the present study, exogenous EE-GRSP applications promoted the accumulation of nutrients, thus improving the cytoplasmic concentration, increasing the osmotic pressure, and decreasing the water potential.

Mineral elements will directly affect the plants’ normal growth, as well as the photosynthetic performance by reducing the chlorophyll content. The level and distribution of nutrient elements in plants also reflect their efficiency in nutrient absorption and utilization [30]. Normally, P has a low mobility and diffusion ability in soil, which is always considered as a limiting element for plant growth [31]. The present study showed that exogenous EE-GRSP application sustainably increased the P content in lemon seedlings, indicating that exogenous EE-GRSP application might have the ability to activate P in soil, thus enhancing the plant absorption and transportation, resulting in the increased utilization of P by plants. In addition, the present study showed a higher Fe content in both the leaves and roots of lemon seedlings under different strengths of exogenous EE-GRSP applications. Interestingly, the exogenous EE-GRSP applications only significantly increased the Mg content in the roots but not in the leaves. It is well known that Mg and Fe are the essential components involved in chlorophyll synthesis [32,33], but chlorophyll contains only Mg but no Fe. In this study, Mg might be more consumed during the chlorophyll synthesis. Meanwhile, photosynthesis is inseparable from chlorophyll, so the content of Mg and Fe may indirectly affect photosynthesis, which also provides a possible explanation for the EE-GRSP promoting plant growth in this study. In addition, the increase in Fe content in lemon seedlings after EE-GRSP application in this study might also be related to the fact that GRSP contains Fe [34]. And these results are also consistent with Liu et al. [20] on trifoliate orange. Furthermore, exogenous EE-GRSP application significantly decreased leaf Zn, root Ca, and Mn in both the leaves and roots, and the effects of these three elements on plants might mainly focus on enhancing plant resistance [35,36,37]. Additionally, exogenous EE-GRSP applications did not significantly affect the root N content, but high-strength EE-GRSP applications dramatically decreased the leaf N content, and it is possible that GRSP only maintained the plants’ N at its required stable level. After all, excessive N concentrations would inhibit plant growth [38].

It is well known that AMF can absorb water from the soil through its widely spread mycelia and transport it to plant roots [39,40]. Approximately 4% of water in plant roots originates from mycorrhizal hyphae [39]. GRSP, a secreted substance of AMF, can serve as a hydrophobic layer and has a pronounced ability to distribute on the surface of soil aggregates, greatly reducing the water loss of soil aggregates [9]. Generally, the expression levels of *TIPs* reflect the water content and absorption capacity of plants [41]. In this study, the results revealed that the EE-GRSP-mediated *TIP* expression was affected by its concentrations or strengths. In detail, 1/4-strength exogenous EE-GRSP applications had the most significant effect on *ClTIP* expressions, manifested as a significant up-regulation of all *ClTIPs* of lemon seedlings. Possibly, 1/4-strength exogenous GRSP application resulted in an increase in soil solution concentration, which caused dehydration stress to lemon seedlings. This promoted an increase in AQP expression in the plant to improve membrane permeability and facilitate water transport in the host plant [42]. The result suggested that low-strength EE-GRSP applications can enhance the membrane permeability of lemon leaves and roots, thereby promoting the water absorption and transport. However, high-strength exogenous EE-GRSP applications dramatically down-regulated most of the *ClTIP* expressions, wherein 1/2- (excepted leaf *ClTIP2;2*) and full-strength exogenous EE-GRSP applications interestingly down-regulated all the *ClTIP* expressions, and 3/4-strength EE-GRSP application significantly down-regulated the expressions of three *ClTIPs* (*ClTIP1;3*, *ClTIP2;1*, *ClTIP4;1*) in the leaves and three (*ClTIP1;2*, *ClTIP2;2*, *ClTIP5;1*) in the roots. The expression of *ClTIPs* was down-regulated under high-strength exogenous EE-GRSP applications, which might be due to a decrease in membrane permeability, thereby keeping plant cells hydrated [42]. On the other hand, higher-strength exogenous EE-GRSP applications may regulate the stability or down-regulation of *TIP* genes to reduce water dissipation and prevent water reflux into dry soil [43]. Although previous research has found that exogenous EE-GRSP applications promoted the root development of trifoliate orange—thus improving the water uptake, which was accompanied by an increased leaf water potential [7]—and the role of AQP expression in this process has not been studied, the involvement of AQPs in regulating plant water absorption under drought stress is needed in subsequent studies, especially when GRSP is applied.

## 4. Material and Methods

### 4.1. Experimental Material

The seeds of the plant material, lemon (*Citrus limon* L.), were provided by the Resear-ch Institute of Fruit and Tea, Hubei Academy of Agricultural Sciences. The method of Liu et al. [44] was referred to regarding seed disinfection and germination, with the germination carried out in sterilized sand (121 °C, 0.1 MPa, 1 h) for 4 weeks under the conditions of 28/20 °C (day/night) and 80% relative humidity. One non-mycorrhizal seedling with two euphylla of uniform size were selected and transplanted into one pot.

EE-GRSP was extracted from the soil collected from an 11-year-old citrus orchard (Yangtze University, Hubei Province, China) at a depth of 0–20 cm. The soil was well mixed and air-dried, and the roots and debris were removed through a 4 mm sieve. EE-GRSP extraction was conducted according to the protocol of Koide and Peoples [45] and Chi et al. [7]. The details were as follows: 4 g of sieved soil sample was extracted with 32 mL of 20 mmol·L^−1^ citrate buffer (pH 7.0) at 0.1 MPa and 121 °C for 30 min, and centrifuged for 5 min at 10,000× *g*/min. The supernatant was collected as the stock solution and defined as a full-strength EE-GRSP solution with a concentration of 0.027 mg protein mL^−1^, which was determined according to Bradford [46] using bovine serum albumin as a standard. And the stock solutions were stored at 4 °C for less than one month.

### 4.2. Experimental Design

A completely randomized design with different strengths of exogenous EE-GRSP as the factor was used. The experiment consisted of five treatments consisting of zero-strength EE-GRSP with citrate buffer (20 mmol, pH 7.0) as the control, and quarter-strength (1/4 EE-GRSP), half-strength (1/2 EE-GRSP), three-quarter-strength (3/4 EE-GRSP), and full-strength (full EE-GRSP). Meanwhile, the 1/4-, 1/2-, and 3/4-strength EE-GRSPs were made by a certain volume ratio of full-strength EE-GRSP solution with 20 mmol/L, pH 7.0, of citric acid buffer. Each treatment was replicated five times, with one replicate per pot, for a total of 25 pots.

One two-leaf-old lemon seedling with uniform size was selected and transplanted into a 2 L plastic pot containing 1.5 kg of autoclaved (121 °C, 0.11 MPa, 2 h) yellow soil. The soil had a pH of 6.1, and organic carbon (C), available N, Oslen-phosphorus (P), and available potassium (K) were 9.7 mg/kg, 11.8 mg/kg, 15.3 mg/kg, and 21.5 mg/kg, respectively. After transplanting, 80 mL of distilled water per pot was irrigated every 2 days for the first 4 weeks to maintain plant growth, and then 80 mL of designed EE-GRSP solution was supplied weekly to each pot for another 4 weeks. During the treatment period, distilled water was supplemented according to the soil moisture status to maintain appropriate water status. After EE-GRSP treatment, the plants were managed routinely for 4 weeks and harvested after a total of 12 weeks.

### 4.3. Variable Determinations

Plant height, stem diameter, and leaf number were recorded before harvest. The seedlings were divided into leaves, stems, and roots, and immediately determined. Then, the whole root systems of each treatment were taken out carefully and subsequently scanned using the Epson PerfectionV700 Photo Dual Lens System (J221A, Seiko Epson Corporation, Jakarta Selatan, Indonesia). The root morphological variables including total length, surface area, and volume were analyzed using the WinRHIZO software (Regent Instruments Inc., Montreal, QC, Canada).

The relative water content (RWC) of leaves was determined according to the method of Sade et al. [47]. The fresh weight (FW) of the lemon leaves was quickly weighed, the leaves were immersed in distilled water for 24 h, and the saturated weight (TW) was weighed, followed by drying of the leaves in an oven at 80 °C until constant weight and weighing the dry weight (DW). The formula is: RWC (%) = (FW − DW)/(TW − DW) × 100%. 

The method for determining leaf water potential was improved on that of Knipling [48]. The uniform leaves were selected and perforated symmetrically along both sides of the main vein with a perforator, and then 10 randomly selected leaf discs were immersed in the test tube, which were filled with different concentrations (0.1, 0.2, 0.3, 0.4, 0.5, 0.6, 0.7, 0.8 mol/L) of sucrose solutions for 30 min and shaken several times, during which time a small amount of methylene blue powder was added to each test tube. Subsequently, a small amount of liquid was sucked from each test tube and inserted into the middle of the solution of the same concentration of sucrose solutions, and a drop was slowly released to observe the rise and fall of the droplet. The water potential was calculated by taking the average of the two solution concentrations.
Calculation formula: ψ_w_ = ψ_s_ = -iCRT

In the formula, ψw is the water potential (in MPa); ψs is the osmotic potential (in MPa); i is the isostatic coefficient of the solution, sucrose, which is 1.0; C is the concentration of the substance of the solution (in mol/kg); R is the molar gas constant (taken as 0.008314 MPa·L· mol^−1^·K^−1^); and T is the thermodynamic temperature (in K).

The roots were treated at 105 °C for 30 min and then oven-dried at 70 °C for 48 hground into 0.5 mm powder. The N content in leaves and roots was determined by a chemical analyzer (Smart Chem 200) according to the method of Parkinson and Allen [49], after the plant samples were extracted using H_2_SO_4_ -H_2_O_2_ solution. The extraction method was as follows: the sample was placed onto the bottom of the digestive tube, and 5 mL of H_2_SO_4_ was stewed overnight. After preheating at 160 °C, the temperature was raised to 350 °C when H_2_SO_4_ emitted white smoke, and then 4–6 drops of H_2_O_2_ were added every 15 min until the solution became colorless. According to the method of Medveckienė et al. [50], other mineral elements such as P, K, calcium (Ca), magnesium (Mg), iron (Fe), copper (Cu), manganese (Mn), zinc (Zn), and B were determined by an Ion Emission Spectrometer (IRIS Advantage) after the leaves and roots of plants were extracted by HNO_3_ HCl, and the operation steps were the same as H_2_SO_4_ -H_2_O_2_.

Total RNA was extracted from lemon roots and leaves (0.2 g) using the TaKaRa kit (TaKaRa MiniBEST Universal RNA Extraction Kit). The RNA concentration and purity were detected in an ultra-micro spectrophotometer (K5600C, Beijing Kaio Technology Development Co., Beijing, China). The RNA with high purity was selected for reverse transcription using TaKaRa kit (PrimeScript™ RT reagent Kit with gDNA Eraser), according to the manufacturer’s instructions. *TIP* family genes were screened based on previous transcriptome data and the citrus database (Citrus Pan-genome2breeding Database (hzau.edu.cn), accessed on 17 January 2021), and the primers of the *TIP* gene and internal reference were designed using Primer Premier 5.0 (Palo Alto, CA, USA) software and synthesized by Shanghai SangonBio. Tech. Co. (Wuhan China) (Table 4). Each reaction mixture for qRT-PCR consisted of 7.2 µL of ddH_2_O, 2 µL of cDNA, 10 µL of AceQ qPCR SYBR Green Master Mix, 0.4 µL of forward primer, and 0.4 µL of reverse primer for a total of 20 µL. Reaction systems were run on a CFX96 real-time PCR detection system (BIO-RAD, Berkeley, CA, USA): 95 °C for 5 min for pre-warming, 40 cycles at 95 °C for 10 s, 60 °C for 30 s and 72 °C for 30 s, and finally ending the run with 95 °C for 15 s, 60 °C for 60 s, and holding at 95 °C for 15 s. The experiment was replicated three times (biological replicates) with three technical replications for each gene. The relative expression of each target gene was calculated using the 2^−∆∆Ct^ methods.

### 4.4. Statistical Analysis

The data were analyzed by one-way analysis of variance using SAS (8.1) statistical analysis software. The significance level was 5%, and Sigma Plot 15.0 was used for plotting.

## 5. Conclusions

The promoting effect of exogenous EE-GRSP applications on the plant growth performance of lemon seedlings was related to its strengths or concentrations, and the responses of exogenous EE-GRSP applications on lemon seedling were associated with the improvement in water status through an optimized root morphology and diverse *ClTIP* expressions. Owing to its particular and complex functions of *TIPs*, the different strengths of exogenous EE-GRSP applications included up-regulations, down-regulations, and no significant regulations of familiar members of root and leaf *TIPs*, indicating that different strengths of EE-GRSP applications have different responses to AQPs.

## Figures and Tables

**Figure 1 plants-12-02955-f001:**
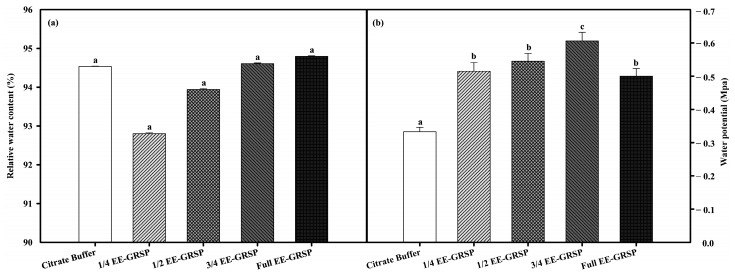
Effects of different strengths of exogenous EE-GRSP on relative water content (**a**) and leaf water potential (**b**) of lemon seedlings (*Citrus limon* L.). Data (means ± SD, *n* = 5) followed by different letters above the bars indicate significant differences (*p* < 0.05) between treatments.

**Figure 2 plants-12-02955-f002:**
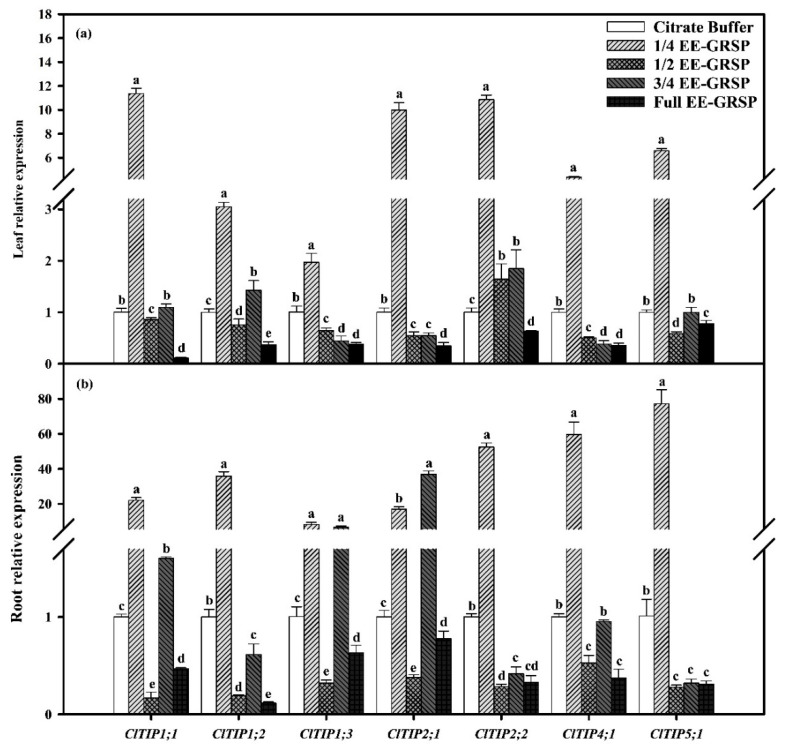
Effects of different strengths of exogenous EE-GRSP on the relative expression of *ClTIPs* in leaves (**a**) and roots (**b**) of lemon seedlings (*Citrus limon* L.). Data (means ± SD, *n* = 5) followed by different letters above the bars indicate significant differences (*p* < 0.05) between treatments.

**Table 1 plants-12-02955-t001:** Effects of different strengths of exogenous EE-GRSP on plant growth performance and biomass of lemon seedlings (*Citrus limon* L.).

Treatment	Plant Height (cm)	Stem Diameter (mm)	Leaf Number (#/plant)	Biomass (g FW/plant)		
				Leaf	Shoot	Root
Citrate buffer	8.42 ± 0.75 c	0.30 ± 0.04 b	8.60 ± 1.14 b	0.39 ± 0.05 d	0.38 ± 0.04 e	1.06 ± 0.07 c
1/4 EE-GRSP	9.40 ± 0.91 b	0.32 ± 0.02 b	9.40 ± 1.52 b	0.57 ± 0.05 c	0.58 ± 0.01 d	0.87 ± 0.09 c
1/2 EE-GRSP	13.60 ± 0.53 a	0.39 ± 0.04 a	13.40 ± 1.92 a	1.22 ± 0.11 b	1.12 ± 0.04 b	2.24 ± 0.30 a
3/4 EE-GRSP	14.32 ± 0.73 a	0.40 ± 0.04 a	13.80 ± 1.67 a	1.07 ± 0.05 b	0.91 ± 0.01 c	1.80 ± 0.14 b
Full EE-GRSP	13.80 ± 0.57 a	0.37 ± 0.03 a	13.80 ± 2.59 a	1.70 ± 0.16 a	1.18 ± 0.05 a	1.95 ± 0.12 ab

Data (means ± SE, *n* = 5) followed by different letters among treatments indicate significant differences at *p* < 0.05.

**Table 2 plants-12-02955-t002:** Effects of different strengths of exogenous EE-GRSP on root system architecture of lemon seedlings (*Citrus limon* L.).

Treatment	Total Length (cm)	Projected Area (cm²)	Surface Area (cm²)	Average Diameter (mm)	Volume (cm³)	Taproot Length (cm)	Lateral Root Numbers (#/plant)		
							1st-Order	2nd-Order	3rd-Order
Citrate Buffer	156.6 ± 9.4 a	11.79 ± 0.65 a	12.20 ± 0.53 c	0.71 ± 0.01 a	0.73 ± 0.02 c	9.33 ± 1.53 c	52 ± 3 c	54 ± 3 d	23 ± 3 a
1/4EE-GRSP	160.3 ± 12.7 a	10.18 ± 0.34 b	12.49 ± 1.05 bc	0.72 ± 0.09 a	0.86 ± 0.09 c	11.83 ± 1.61 b	43 ± 3 c	63 ± 7 d	12 ± 3 b
1/2EE-GRSP	164.5 ± 6.6 a	11.47 ± 0.68 ab	13.19 ± 0.43 bc	0.71 ± 0.05 a	1.61 ± 0.15 a	16.67 ± 0.58 a	80 ± 9 ab	223 ± 12 b	13 ± 3 b
3/4EE-GRSP	180.7 ± 14.9 a	12.82 ± 1.13 a	13.97 ± 0.33 a	0.65 ± 0.06 a	1.37 ± 0.08 b	18.67 ± 1.53 a	75 ± 7 b	155 ± 12 c	12 ± 4 b
Full EE-GRSP	168.5 ± 15.2 a	12.34 ± 0.56 a	13.58 ± 0.46 ab	0.64 ± 0.03 a	1.40 ± 0.04 b	12.33 ± 1.15 b	86 ± 4 a	256 ± 13 a	19 ± 2 a

Data (means ± SE, *n* = 5) followed by different letters among treatments indicate significant differences at *p* < 0.05.

**Table 3 plants-12-02955-t003:** Effects of different strengths of exogenous EE-GRSP on mineral element content in lemon seedlings (*Citrus limon* L.).

Treatment	N (g/kg)	P (g/kg)	K (g/kg)	Ca (g/kg)	Mg (g/kg)	Fe (g/kg)	Mn (mg/kg)	Zn (mg/kg)
Leaf								
Citrate Buffer	16.91 ± 0.32 a	1.15 ± 0.09 e	14.72 ± 1.20 a	22.32 ± 0.53 a	2.74 ± 0.21 a	0.11 ± 0.01 d	26.74 ± 1.45 a	0.25 ± 0.02 a
1/4 EE-GRSP	17.16 ± 0.42 a	1.96 ± 0.12 d	15.21 ± 1.62 a	21.95 ± 0.89 a	2.89 ± 0.15 a	0.14 ± 0.01 c	19.20 ± 1.33 bc	0.17 ± 0.00 c
1/2 EE-GRSP	8.88 ± 0.37 b	3.94 ± 0.09 b	16.24 ± 1.62 a	21.76 ± 0.48 a	2.80 ± 0.17 a	0.14 ± 0.01 c	21.00 ± 0.66 b	0.19 ± 0.01 b
3/4 EE-GRSP	9.37 ± 0.19 b	5.05 ± 0.08 a	15.91 ± 1.48 a	22.00 ± 1.07 a	3.07 ± 0.23 a	0.21 ± 0.01 a	18.35 ± 1.13 c	0.15 ± 0.01 cd
Full EE-GRSP	9.51 ± 1.18 b	3.62 ± 0.15 c	16.83 ± 1.27 a	21.70 ± 0.72 a	2.77 ± 0.19 a	0.19 ± 0.01 b	18.77 ± 0.23 c	0.14 ± 0.00 d
Root								
Citrate Buffer	5.31 ± 0.35 a	1.92 ± 0.06 d	10.76 ± 0.59 a	14.13 ± 0.19 a	1.56 ± 0.05 c	5.27 ± 0.25 d	140.46 ± 1.37 a	0.40 ± 0.02 d
1/4 EE-GRSP	5.06 ± 0.14 ab	1.96 ± 0.25 d	9.76 ± 0.71 a	11.59 ± 0.42 b	1.64 ± 0.06 c	5.59 ± 0.04 d	105.56 ± 5.17 b	0.43 ± 0.02 cd
1/2 EE-GRSP	4.83 ± 0.10 b	2.51 ± 0.02 b	10.28 ± 0.61 a	10.71 ± 0.42 c	2.22 ± 0.13 ab	6.15 ± 0.24 c	85.27 ± 1.17 c	0.86 ± 0.01 b
3/4 EE-GRSP	4.89 ± 0.07 b	2.79 ± 0.01 a	11.04 ± 0.17 a	11.70 ± 0.40 b	2.42 ± 0.16 a	9.44 ± 0.50 a	65.50 ± 0.71 d	1.00 ± 0.10 a
Full EE-GRSP	5.03 ± 0.09 ab	2.20 ± 0.03 c	9.79 ± 1.13 a	10.05 ± 0.07 d	2.13 ± 0.11 b	7.13 ± 0.13 b	56.58 ± 0.85 e	0.49 ± 0.02 c

Data (means ± SE, *n* = 5) followed by different letters among treatments indicate significant differences at *p* < 0.05.

**Table 4 plants-12-02955-t004:** Gene-specific primer sequences used in this study.

Genes	Gene ID	Sequence of Forward Primer (5′→3′)	Sequence of Reverse Primer (5′→3′)
*ClTIP 2;1*	Cs1g15440.1	CCTTCAAGGCCTATCTTGCTGAGTT	CCTGATGGATCAAGTGCTGCATCTG
*ClTIP 4;1*	Cs4g19580	TCTTCATCGTCAGGTCTCATGTTCC	ATAGTGACCCACAGATAACACCGTG
*ClTIP 2;2*	Cs5g08710.1	TTGAATGCCGCTGAAGGTTTGGTTA	AGCAATGGGCGCAATTGTTCCTAGT
*ClTIP 1;3*	Cs7g28650	GATCGTGGAGAGAATTGAAGTTGTG	GCAAGACATAAATCCATCCACTCCT
*ClTIP 1;2*	Cs8g17900	TAAAAAATGCCCGGAATTGCTATCG	TCCATTGTCCGTGAGCTTGCTGTAA
*ClTIP 5;1*	Cs9g14770	AGTCCAGACGCAGCATCAAATACAT	ACGGTGATGTGTCCACTCACTGCCT
*ClTIP 1;1*	Orange1.1t03005.1	TCGCTAATCCACTGCTAACCAACTT	TGGTCTCCACTTCTCAACTAGGAGC
*β-Actin*	Cs1g05000	CCGACCGTATGAGCAAGGAAA	TTCCTGTGGACAATGGATGGA

## Data Availability

All the data supporting the findings of this study are included in this article.

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
