# Peer review of "Exogenous Easily Extractable Glomalin-Related Soil Protein Stimulates Plant Growth by Regulating Tonoplast Intrinsic Protein Expression in Lemon"

_plants, 2023, doi:10.3390/plants12162955_

Round 1

Reviewer 1 Report

The study follows in the line of similar studes conducted by the research team, and hence the study is not entirely novel. My main concerns with the approach are two-fold. Firstly, how can you be certain that there was not another growth stimulating factor (or factors) in your soil extract apart from EE-GRSP? What if the unidentified factor is causing the observed results and not the EE-GRSP? Secondly, you claim that the EE-GRSP is improving water relationships and drought resistance but you have not imposed water stress. Therefore these claims are not supported by your data.

Some other issues that need addressing:

L164/165. This statement is not support by your data! Table 3 column 3 does not show a significant change in projected area!

L176/177 poorly worded, rephrase.

Figure 1, caption for panel a/b swapped around. Alos the scale/unit for panel is incorrect ? It shows %. Please check.

The Discussion section is a bit "overcooked" regarding the effect of increased nutrient uptake on photosynthesis. You have not measured photosynthesis to make these claims.

I also suggest that you briefly summarise the methods used rather than referring to another paper. Have these methods really been used without any modifications? If this is indeed the case, please state so.

Only minor spelling errors found, occasional incorrect word usage, but overall clearly written and easy to follow. Requires only minor language editing.

Author Response

Dear reviewer, thank you very much for all the work you have done for our article. According to your comments, we have made a lot of modifications to the revised paper. Please refer to and give further guidance for the revised manuscript. According the your comments, we have given the response explain “point by point”. Please check it. Thank you very much.

Reviewer 2 Report

MS titled "Exogenous easily extractable glomalin-related soil protein stim- ulates plant growth via promoting water status by regulating 3 TIPs expression in lemon" provides interesting information about the positive effect of soil isolate on growth performance and expression of selected genes related to water utilization in lemon seedlings. I miss the analysis of the soil extract. Authors should provide some proof of glomalin-related soil protein (gel electrophoresis or some other proof). If the applied extract contains some other components you should change the title. This experimental design presented treatments on seedlings grown in optimal water conditions. It would be desirable to present data under drought conditions.

All my suggestions are in attached file.

Author Response

(The authors gave the same response as above.)

Reviewer 3 Report

the effects  on plant growth,root system and the changes of mineral elements contents and relative expression of tonoplast intrinsic protein genes in lemon seedlings were investigated under exogenous EE-GRSP treated, The results showed that exogenous EE-GRSP exhibited differential responses on plant growth which can be used as a biological promoter in plant cultivation. the experiment material part is too concise , the reference author should be added, and the references number need to concised.

the quality of English language is fine, which can be accepted.

Author Response

(The authors gave the same response as above.)

Round 2

Reviewer 1 Report

The researchers have improved the paper. The issues I have raised have been addressed in sufficient detail.

The researchers have improved the paper. It still has some minor language issues which need to be fixed but these could be fixed during the editorial stage.

Reviewer 2 Report

I can see that the authors revised certain parts of MS. In addition, they explained what was unclear to me in the original version of MS.

I can recommend acceptance of this MS for publication.